# Insulin-Lowering Diets in Metastatic Cancer

**DOI:** 10.3390/nu14173542

**Published:** 2022-08-27

**Authors:** Sherry Shen, Neil M. Iyengar

**Affiliations:** 1Memorial Sloan Kettering Cancer Center, New York, NY 10065, USA; 2Weill Cornell Medical Center, New York, NY 10065, USA

**Keywords:** insulin, diet, metastatic cancer

## Abstract

Hyperinsulinemia is an independent risk factor for cancer mortality. Insulin-lowering dietary strategies such as calorie restriction (CR), low-carbohydrate or ketogenic diets (KD), and intermittent fasting (IF) are aimed at reducing systemic stores of nutrients utilized by cancer cells, attenuating insulin-related growth signaling, and improving obesity-related metabolic parameters. In this narrative review, we searched the published literature for studies that tested various insulin-lowering diets in metastatic cancer in preclinical and clinical settings. A total of 23 studies were identified. Of these, 14 were preclinical studies of dietary strategies that demonstrated improvements in insulin levels, inhibition of metastasis, and/or reduction in metastatic disease burden in animal models. The remaining nine clinical studies tested carbohydrate restriction, KD, or IF strategies which appear to be safe and feasible in patients with metastatic cancer. These approaches have also been shown to improve serum insulin and other metabolic parameters. Though promising, the anti-cancer efficacy of these interventions, such as impact on tumor response, disease-specific-, and overall survival, have not yet been conclusively demonstrated. Studies that are adequately powered to evaluate whether insulin-lowering diets improve cancer outcomes are warranted.

## 1. Introduction

In 2022, there will be an estimated 1.9 million new diagnoses of cancer and over 600,000 cancer deaths in the United States [1]. Hyperinsulinemia is a well-established independent risk factor for the development of many types of cancers as well as cancer mortality [2,3,4,5,6,7,8,9,10,11,12,13,14,15,16]. In fact, the proportion of cancers with attributable risk to high body mass index and diabetes is estimated at 31% of endometrial cancers, 29% of esophageal cancers, 18–21% of renal cancers, 16–18% of liver cancers, 13% of pancreatic cancers, and 7% of breast cancers [17]. While advances in cancer therapeutics have resulted in decreasing cancer mortality rates, there is growing evidence that metabolic dysfunction limits the efficacy of cancer treatment and can in fact be exacerbated by it, thereby establishing a paradoxical cycle of metabolic perturbation and treatment resistance (Figure 1).

Various diets that modulate insulin levels have been proposed as adjunctive therapies to cancer treatment, including calorie restriction, low-carbohydrate and ketogenic diets, and intermittent fasting strategies. These dietary strategies have demonstrated efficacy in reducing disease progression and metastasis formation in preclinical studies. There are few randomized clinical studies in patients with metastatic cancer, partly due to concerns regarding pre-existing cachexia, sarcopenia, and muscle wasting that might be exacerbated by caloric restriction. However, even in these settings, hyperinsulinemia can be present, as in sarcopenic obesity [18]. Thus, precision nutrition trials are needed in the metastatic setting that aim to improve metabolic profiles, body composition, and cancer outcomes without significant weight reduction or other safety concerns.

In this review, we describe the proposed mechanisms of insulin-lowering diets in countering tumor progression and present preclinical evidence supporting further clinical studies of insulin-lowering diets to reduce metastatic disease burden. We present preliminary clinical studies that address feasibility and implementation of insulin-lowering diets in patients with metastatic cancer and the potential impact of these diets on cancer outcomes. Finally, we discuss ongoing and future clinical trials and provide recommendations for best practices in designing trials utilizing insulin-lowering dietary strategies in patients with metastatic cancer.

## 2. Materials and Methods

We conducted a PubMed search using variants of the following keywords: “calorie restriction”, “ketogenic diet”, “low-carbohydrate diet”, “intermittent fasting”, “metastatic cancer”, with restriction to English language. For clinical studies, we included observational studies and clinical trials studying the effects of various insulin-lowering diets in patients with metastatic cancer; we excluded case reports. We reviewed references lists of identified articles to find additional publications.

## 3. Review

### 3.1. Insulin-Lowering Diets as an Anti-Cancer Therapeutic Strategy

One of the hallmarks of cancer is altered energy metabolism, which promotes enhanced cell proliferation [19]. Normal cells, under aerobic conditions, utilize glycolysis to convert glucose to pyruvate in the cytoplasm and subsequently to carbon dioxide in mitochondria. Cancer cells exhibit altered glucose metabolism, relying on glycolysis for energy production rather than oxidative phosphorylation even in the presence of oxygen and functional mitochondria, known as the Warburg effect [20]. To support this metabolic adaptation, an abundance of nutrients including sugars and fatty acids is necessary, which are typically acquired via dietary intake and/or energy stores such as adipose tissue.

Another metabolic adaptation that occurs in cancer cells involves the insulin signaling pathway. Binding of insulin to the insulin receptor (IR) induces a conformational change in the receptor to exert tissue-specific metabolic effects. Effector proteins include phosphoinositide 3-kinase (PI3K), which in conjunction with mammalian target of rapamycin complex 2 (mTORC2) activates AKT, resulting in cellular glucose uptake and inhibition of gluconeogenesis, lipolysis, and apoptosis and promotion of glycogen synthesis, lipogenesis, and cell proliferation [21]. Activation of the rat sarcoma-mitogen-activated protein kinase/ERK (Ras-MAPK/ERK) signaling pathway also promotes cell growth, survival, and differentiation. Cancer cells often overexpress the insulin receptor with a preferential expression of the IR-A isoform, which promotes growth and survival; knockdown in cancer cell lines results in smaller tumors and fewer metastases [22,23,24,25,26]. In addition to these directly mitogenic effects, hyperinsulinemia affects sex hormone levels which can directly stimulate hormonally driven cancers such as breast and endometrial cancers [27,28]. Clinically, hyperinsulinemia and other obesity-related factors increase the risk of developing several different types of cancers and can lead to increased cancer-specific mortality [29,30,31].

Insulin-lowering diets target these metabolic adaptations. There are three key pathways through which insulin-lowering diets can reduce tumor volume and metastasis formation: (1) by directly reducing systemic stores of nutrients that are utilized by cancer cells, (2) by altering insulin-related signaling factors which stimulate tumor growth, and (3) by improving obesity-related metabolic parameters such as adiposity, dyslipidemia, and systemic inflammation [32]. Various types of insulin-lowering diets have been studied and can be broadly categorized as modulating dietary content, adjusting timing of intake, or both. Specific dietary interventions that have been tested in the metastatic setting include calorie restriction (CR), ketogenic diet (KD), low-carbohydrate diet (LCD), and intermittent fasting (IF).

CR consists of a lower total daily calorie intake than the standard diet but maintains a balanced macronutrient ratio. Implementation of a 25% CR diet in sedentary but otherwise healthy adults leads to weight loss and improved fasting insulin levels and use of a CR diet in overweight or obese patients with diabetes additionally leads to improvements in hemoglobin A1c, lipid profile parameters, and systolic blood pressure [33,34]. While CR has been the most widely studied in preclinical models of metastatic cancer compared with the other dietary strategies (reviewed in detail in Section 3.3), it has not been utilized to the same degree in patients with metastatic cancer due to concerns regarding weight loss and cachexia.

Two dietary strategies, KD and LCD, maintain the standard daily caloric intake but with adjustment of macronutrient ratios. The ketogenic diet is a very low carbohydrate diet consisting of 30–40 g of carbohydrates per day, whereas a low-carbohydrate diet consists of less than 100 g of carbohydrates per day. These contrast with the typical Western diet, which contains an average of 200–300 g of carbohydrates per day, and thus represent a significant dietary alteration [35]. The premise of both of these diets is to reduce total dietary carbohydrate intake and liver glycogen stores. Additionally, the very low carbohydrate content of the KD induces ketogenesis as a source of energy. These diets can lead to weight loss, decreased adiposity, decreased fasting glucose and insulin levels, improved insulin sensitivity, and improved blood pressure [36,37].

IF strategies encompass many types of approaches that alter the timing of dietary intake. Two types of IF diets that have been studied in preclinical settings with effects on cancer metastasis are time-restricted feeding (TRF) and fasting-mimicking diet (FMD). TRF modifies timing by limiting food intake to specified daytime intervals ranging 6–12 h and prolongs the period of fasting during nocturnal hours. Fasting-mimicking diets (FMD), on the other hand, modify both timing and macronutrient intake to mitigate hunger and other challenges of prolonged fasting, where significant caloric restriction (limited to 35–55% of usual intake) occurs for 5–7 consecutive days per month while intake is not restricted for the rest of the days in the month [38]. These diets do not typically impact weight to the degree that the content-modulating diets described above do, but do improve body fat percentage, waist circumference, fasting insulin, insulin-like growth factor 1 (IGF1), and leptin levels [39,40,41].

### 3.2. Included Studies

Based on our search methods, a total of 23 studies were identified. Of these, 14 were preclinical studies of dietary strategies that demonstrated improvements in insulin levels, inhibition of metastasis, and/or reduction of metastatic disease burden in animal models. There were 9 clinical studies that tested KD, LCD, or IF strategies in patients with metastatic cancer. We summarize and discuss the preclinical studies in Section 3.3 and the clinical studies in Section 3.4. The effects of each of these diets in the preclinical and clinical settings are summarized in Figure 2.

### 3.3. The Effects of Insulin-Lowering Diets on Metastasis in Animal Models

Studies describing the effects of insulin-lowering diets on metastasis in the preclinical setting are described in Table 1. Various preclinical studies have demonstrated the benefits of CR in metastatic cancer. In orthotopic mammary tumor models, 40% CR compared to no restriction resulted in reduced growth of metastatic tumors and decreased metastasis formation, both of metastases that originate spontaneously from the primary tumor and experimentally from intravenous injection of 4T1 mammary cancer cells [42]. Calorie-restricted mice had lower levels of insulin, IGF1, leptin, and increased adiponectin. Additionally, lower levels of various factors such as transforming growth factor-β, intratumor collagen IV, active matrix metalloproteinase 9, and total collagen intratumor volume fraction in the energy-restricted mice indicate systemic metabolic changes that affected the stroma and tumor cells, creating a microenvironment that reduces tumor growth and metastasis [42]. CR by 30% in an orthotopic syngeneic triple-negative breast cancer model demonstrated significant reduction in tumor growth rate and number of metastatic lesions compared with controls [43]. Another study in an isogeneic mouse model of ovarian cancer also utilized 30% CR and showed that mice given this diet had decreased metastatic tumor burden as measured by number and size of tumor nodules and lower ascites volume as compared with a regular diet or high-energy diet [44].

Several preclinical studies have examined a KD in the metastatic setting. Administration of a KD in the firefly luciferase-tagged VM-M3 mouse model of metastatic cancer demonstrated significantly slowed tumor growth and increased survival [45]. In a study of nude mice injected with a gastric adenocarcinoma cell line, mice fed a KD had significantly delayed tumor growth compared with those fed a standard diet [46]. In a separate study, mice inoculated with the 4T1 breast cancer cell line were given a KD or control diet; those that consumed the KD had lower levels of circulating tumor cells, which correlated with lower metastatic disease burden [47]. Although not induced by diet, exogenous ketone supplementation with either 1,3-butanediol or a ketone ester decreased proliferation and viability of cultured cancer cells in vitro and prolonged survival in vivo in mice implanted with firefly luciferase-tagged syngeneic VM-M3 cells (a metastatic cancer model), independent of glucose levels [48].

Use of a low-carbohydrate, high-protein diet in a mouse prostate cancer model with predisposition to metastases resulted in a significantly decreased incidence of metastasis formation as compared with a diet containing standard Western macronutrient ratios. Notably, “low-carbohydrate” with 15% of the carbohydrate content of a standard Western diet used in this study aligns with the typical carbohydrate content of a ketogenic diet. Two of the eight mice in the 15% carbohydrate group died of non-cancer causes which may suggest that severe carbohydrate restriction could be deleterious [49]. While this safety signal has not been borne out in studies of KD or LCD in men with early stage prostate cancer, these dietary strategies have not been tested for safety specifically in the metastatic prostate cancer setting [50,51].

**Table 1 nutrients-14-03542-t001:** Summary of preclinical studies testing insulin-lowering dietary strategies in models of metastatic cancer.

Author	Year	Model	Intervention	Findings
De Lorenzo [42]	2011	Orthotopic 4T1 mammary cancer	40% CR vs. no restriction	↓metastatic tumor growth, ↓metastasis formation, ↓insulin, ↓IGF1, ↓leptin, ↑adiponectin
Phoenix [43]	2010	Orthotopic syngeneic 66cl4 triple-negative breast cancer	30% CR vs. standard diet vs. diet with high levels of free sugar	↓tumor growth, ↓number of metastatic lesions
Al-Wahab [44]	2014	Isogeneic ID8 ovarian cancer	30% CR vs. regular diet vs. high-energy diet	↓metastatic tumor burden, ↓size of tumor nodules, ↓ascites volume
Poff [45]	2013	Syngeneic VM-M3 metastatic cancer	KD vs. standard diet	↓tumor growth, ↑survival
Otto [46]	2008	Xenograft 23132/87 gastric adenocarcinoma	KD vs. standard diet	Delayed tumor growth
Wang [47]	2021	Xenograft 4T1 breast cancer	KD vs. control diet	↓circulating tumor cells, ↓metastatic disease burden
Poff [48]	2014	Syngeneic VM-M3 metastatic cancer	Standard diet supplemented with 1,3-butanediol or a ketone ester vs. standard diet alone	↓proliferation and viability of cells in vitro, ↑survival
Ho [49]	2014	Transgenic prostate cancer (TRAMP)	15% carbohydrate content of standard Western diet vs. standard diet	↓incidence of metastasis formation
Chen [52]	2012	Xenograft A549 lung, HepG-2 liver, or SKOV-3 ovarian	1-day fasting/6-day refeeding cycles × 4 weeks vs. control diet	↑rate of complete tumor regression, ↑survival
Das [53]	2021	Orthotopic Py230 breast cancer	TRF diet vs. ad libitum-fed diet	↓tumor cell proliferation, ↓tumor vascularization, ↓tumor growth, ↓lung metastases
Bonorden [54]	2009	Transgenic prostate cancer (TRAMP)	Intermittent CR (50% consumption × 2 weeks, ad libitum consumption × 2 weeks) vs. ad libitum diet vs. continuous CR	↑latency period prior to tumor growth/detection, ↑survival
Kusuoka [55]	2018	Syngeneic CT26 colon cancer	Continuous CR vs. periodic 1-day fasting/6-day refeeding × 4 weeks vs.	↓cancer stem cells in tumor/blood, ↓tumor weight/metastasis
Simone [56]	2018	Orthotopic 4T1 breast cancer	CR + chemotherapy vs. ad libitum diet + chemotherapy	↓lung metastases, ↑survival
Zuo [57]	2022	Xenograft MCF7-ESR1 breast cancer	FMD + fulvestrant vs. control diet + fulvestrant	↓metastatic disease burden, ↓visible liver metastases

Abbreviations: CR, calorie restriction; IGF1, insulin growth factor-1; KD, ketogenic diet; TRF, time-restricted feeding; FMD, fasting-mimicking diet; ↓, decreased; ↑, increased.

IF strategies have also been tested. In a study by Chen and colleagues, lung, liver, or ovarian tumor-bearing mice that were subjected to periodic 1-day fasting/6-day refeeding cycles for 4 weeks had a 44% rate of complete regression of progressive tumors and metastases and higher survival rates compared to mice fed control diets that had continuous tumor progression and metastasis [52]. In diet-induced obese oopherectomized mice bearing orthotopic breast cancer cells, use of a TRF diet resulted in decreased tumor cell proliferation characterized by Ki67 staining, reduced tumor vascularization, reduced tumor growth, and decreased lung macrometastases compared with ad libitum-fed mice [53].

Head-to-head comparison of different dietary strategies have yielded mixed results. In a transgenic prostate adenocarcinoma mouse model, intermittent CR with 2 weeks of 50% consumption followed by 2 weeks of ad libitum consumption led to an increased latency period prior to tumor growth and detection and improved survival compared with ad libitum diet or continuous CR diet [54]. In contrast, in BALB/c mice injected with CT26 colon cancer cells that were subjected to periodic 1-day fasting/6-day refeeding cycles for 4 weeks, on the day after starvation overconsumption was detected and was associated with an increased number of cancer stem cells in the tumor and blood. Mice given the periodic fasting diet also had increased tumor weight and metastasis; this was not seen with mice subjected to continuous CR [55]. Another study compared continuous CR to a FMD or an isocaloric standard diet in BALB/cJ mice orthotopically implanted with 4T1 breast cancer cells; continuous CR, compared to the other two diets, resulted in markedly fewer metastatic lung nodules.

Use of insulin-lowering diets can also improve anti-cancer treatment efficacy in preclinical models. For example, CR in combination with chemotherapy in 4T1 triple-negative breast tumor bearing mice resulted in downregulation of IGF-1R and IRS signaling, reduced chemotherapy-induced increases in TNF-α and IL-1β, resulted in fewer lung metastases, and improved overall survival compared with mice treated with chemotherapy and fed ad libitum [56]. In ovariectomized immunodeficient female mice injected with MCF7-ESR1 breast cancer cells constituting an estrogen receptor-positive metastatic breast cancer model, growth of metastatic breast cancer cells on liver extracellular matrix increased glucose dependence of the tumor cells in vitro and glucose metabolism pathways were upregulated in vivo [57]. The authors noted that tumor metastatic burden correlated with dietary carbohydrate levels, and found that fulvestrant significantly increased glycolysis and glycogenesis metabolism pathways, potentially explaining the lack of efficacy of fulvestrant in reducing liver metastasis progression in this model. A FMD with concurrent fulvestrant treatment resulted in a synergistic effect to reduce metastatic disease burden, as measured by bioluminescence of tumors, and decreased the number of visible liver metastases.

In summary, each of the insulin-lowering diets CR, KD, and IF strategies have shown benefit in preclinical models in reducing formation of metastases or decreasing metastatic disease burden. However, comparisons of these diets have yielded conflicting results, and point toward continuous or intermittent CR as a potentially more effective strategy when compared with IF.

### 3.4. The Effects of Insulin-Lowering Diets in Patients with Metastatic Cancer

Few studies testing insulin-lowering diets have included patients with advanced or metastatic cancer; these studies are summarized in Table 2.

Two randomized controlled trials included patients with advanced or metastatic disease. In a randomized controlled trial of 80 patients with breast cancer undergoing chemotherapy, patients were randomized to either KD or control for 12 weeks. The KD group had significantly lower serum lactate, alkaline phosphatase, fasting glucose, and insulin levels after 12 weeks as well as greater reduction in body mass index, body weight, and total body fat percentage [58,59]. There was also a greater reduction in tumor size in the KD group (2.7 cm vs. 0.6 cm, *p* < 0.01), although this was measured on sonography or taken from postoperative pathology reports and standard response criteria (e.g., RECIST) were not utilized. At 6 weeks, those in the KD group had higher global quality of life and physical activity scores, but there were no significant differences seen at the conclusion of the 12 week intervention period [60]. Adherence to KD, defined as serum beta-hydroxybutyrate level >0.5, ranged from 66.7–79.2%. In the Carbohydrate and Prostate Study 1 (CAPS1) study, patients with prostate cancer of any stage initiating androgen deprivation therapy were randomized to a low-carbohydrate diet (LCD) plus walking intervention vs. usual diet and exercise. At 3 months, those in the LCD/walking group had lost more weight, had reduced fat mass, lower percent body fat, improved insulin resistance, lower hemoglobin A1c, higher high-density lipoprotein (HDL), and lower triglycerides compared to those in the control group [51]. The diet and exercise intervention also enhanced reduction of androsterone sulfate by androgen deprivation therapy [50]. However, there were no significant differences in prostate-specific antigen (PSA) levels, suggesting no significant difference in disease burden between the arms; while the authors did not specify the proportion of men in each arm with metastatic disease, baseline PSA levels were elevated in the range of 18–21 ng/mL and were similar between treatment arms.

Three single-arm feasibility and safety studies evaluated carbohydrate restriction or KD in patients with advanced cancer of any primary site. In a study of 17 patients with advanced cancer who consumed diets consisting of 20–40 g of carbohydrates per day for 16 weeks, mean weight loss was 12.3 kg (13% of baseline body weight) and quality of life functional and symptom scores improved. Despite this, serum parameters including blood glucose and lipid profile values remained stable, with only HDL cholesterol improving numerically (*p* = 0.06) [61]. Notably, the percentage body weight reduction was significantly greater among patients who achieved partial response or stable disease as compared with patients who had disease progression (*p* = 0.03). In a pilot study of 10 patients with advanced cancers who consumed insulin-lowering diets defined as dietary carbohydrate restriction to 5% of total caloric intake, mean caloric intake decreased by 35% and weight decreased by a median of 4% [62]. Patients who achieved higher diet-induced ketosis as measured by increase in serum β-hydroxybutyrate levels were more likely to have partial response or stable disease. A third study of 16 patients with advanced cancer tested a ketogenic diet for 3 months [63]. However, only 5 patients were able to adhere to the KD; 1 did not tolerate the diet within days, 2 patients died before the end of the intervention period, 1 stopped for personal reasons, 1 was unable to adhere to the diet, 3 discontinued due to progression of disease, and 1 discontinued due to the need to resume chemotherapy. Among the 5 patients who completed the intervention, there were improvements in emotional functioning and insomnia while global quality of life scores remained stable. However, physical function, cognitive function, and symptoms including fatigue, pain, and dyspnea slightly worsened. These three studies demonstrate that carbohydrate restriction is generally feasible, safe, and may improve some metabolic measures in patients with advanced cancers; however, a more restrictive KD strategy may result in difficulties with adherence and affect quality of life. The impact of these interventions on disease burden or response to therapy is difficult to assess in these small sample sizes with heterogeneous populations.

Three studies have evaluated fasting strategies. A study of 20 patients receiving platinum-based chemotherapy included 6 patients with metastatic urothelial, uterine, ovarian, or non-small cell lung carcinomas and demonstrated that short-term fasting for less than 72 h prior to chemotherapy was feasible and safe [64]. Study participants had decreased IGF1 levels after fasting and there was a nonsignificant trend toward less grade 3/4 neutropenia in the cohorts that fasted for 48 or 72 h compared with 24 h. In two ongoing studies NCT03595540 and NCT03340935 testing the safety and feasibility of FMD in cancer patients in conjunction with any type of anti-cancer therapy, a total of 12 patients with metastatic hormone receptor-positive breast cancer have been included (5 in NCT03595540, 7 in NCT03340935). Preliminary results from these patients were reported as part of a study demonstrating the effects of FMD on enhancing endocrine therapy in a hormone receptor-positive mouse model [41]. Patients maintained a stable body weight with increased fat-free mass and decreased total fat mass over time. Data on prior lines of therapy for these patients were not reported, but 11/12 patients derived clinical benefit defined as stable disease, partial response, or complete response. Full results of these studies have yet to be reported.

Finally, two retrospective studies from the same group have examined metabolically supported chemotherapy, defined as fasting- and insulin-induced hypoglycemia followed by administration of chemotherapy, in combination with KD, local hyperthermia to ≥42 °C, and hyperbaric oxygen therapy. The rationale for this type of therapy is to increase acute metabolic stress on cancer cells (via hypoglycemia and KD) and increase efficacy of chemotherapy through increased membrane permeability (via hyperthermia and hyperbaric oxygen therapy, which increases oxygenation of tumor cells). In 44 patients with metastatic non-small cell lung cancer receiving weekly carboplatin/paclitaxel, there was acceptable toxicity and no adverse effects related to hypoglycemia, fasting, or the ketogenic diet [65]. In 25 patients with metastatic pancreatic cancer receiving FOLFIRINOX or gemcitabine-based chemotherapy, there were similarly no adverse effects or toxicities related to fasting, hypoglycemia, or the KD [66]. Median progression-free survival was 12.9 months and medial overall survival was 15.8 months, but these are difficult to interpret in a retrospective study without a control group.

## 4. Discussion

Various insulin-lowering dietary strategies have been tested preclinically and clinically, showing improvements in metabolic factors that are associated with cancer mortality; however, improvement in cancer-specific endpoints have not been demonstrated and thus these strategies cannot yet be recommended as part of routine cancer care in patients with metastatic disease. In preclinical studies, CR, KD, LCD, and IF (TRF and FMD) have all been tested with some degree of success in improving metabolic parameters and decreased metastatic disease burden, although head-to-head comparisons between strategies have yielded mixed results. Clinical studies have focused on varying degrees of carbohydrate restriction, KD, and IF strategies. Carbohydrate restriction and short-interval fasting (24–72 h prior to chemotherapy or FMD) appear to be safe, feasible, and show promising results in terms of improving serum factors such as glucose and insulin and body composition. Two studies suggested potential benefit in disease control, although both included fewer than 20 patients.

There are many challenges to studying insulin-lowering diets in patients with metastatic cancer including poor tolerability, difficulty with adherence to restrictive intake patterns, disease progression and a more frequent need for change in treatment regimens, concerns surrounding weight loss and cachexia, and pre-existing high symptom burden. Several questions remain to be answered: Do insulin-lowering strategies ultimately improve outcomes and prolong survival in patients with metastatic cancer? If so, what are the most effective insulin-lowering strategies? How can these strategies be employed in a manner that is feasible and tolerable for patients with metastatic cancer? How can insulin-lowering diets augment anti-cancer treatments while minimizing adverse effects? Clearly, future studies are needed to address these important questions, yet safely conducting these studies and recruiting eligible patients can be challenging. Maintaining total caloric intake to prevent weight loss while improving systemic insulin levels, metabolic factors, and hormone levels is likely to promote safer implementation, increased adherence, and potentially improvement of metastatic cancer outcomes.

## 5. Future Directions

Several ongoing studies are testing insulin-lowering diet strategies in patients with metastatic cancer, summarized in Table 3. Most of these studies test various degrees of carbohydrate restriction in conjunction with standard anti-cancer therapies or short-term fasting similar to FMD in the peri-infusion period, whether chemotherapy or immunotherapy. Primary endpoints vary but nearly all studies include disease responses, as measured by overall response rate or progression-free survival, as secondary endpoints, as well as quality-of-life assessments. Notably, anticipated sample sizes for these studies are higher than the patient cohorts that are currently reported in the literature. Overall, these trials are anticipated to yield additional important information on feasibility and safety of various insulin-lowering dietary strategies and will hopefully begin to elucidate whether these diets can improve treatment response rates and survival.

To address current gaps in knowledge, future clinical trials should incorporate several considerations: (1) biomarker analysis: identification of biomarkers that can be used to select patients most likely to benefit from insulin-lowering dietary strategies such as somatic mutations involving the insulin signaling pathway (e.g., *PIK3CA* or *AKT*); (2) safety parameters: inclusion of patients with elevated body mass index or fat mass, normal muscle mass, and/or exclusion of patients with cachexia or failure to thrive syndromes; (3) appropriate endpoints: prioritization of serum metabolic markers that are associated with outcomes and inclusion of disease-related endpoints. Dietary strategies as part of anticancer therapy are of great interest to patients and are supported by preclinical data; with a precision approach to clinical trial design, an impactful role for these strategies may be identified in patients with metastatic cancer.

## Figures and Tables

**Figure 1 nutrients-14-03542-f001:**
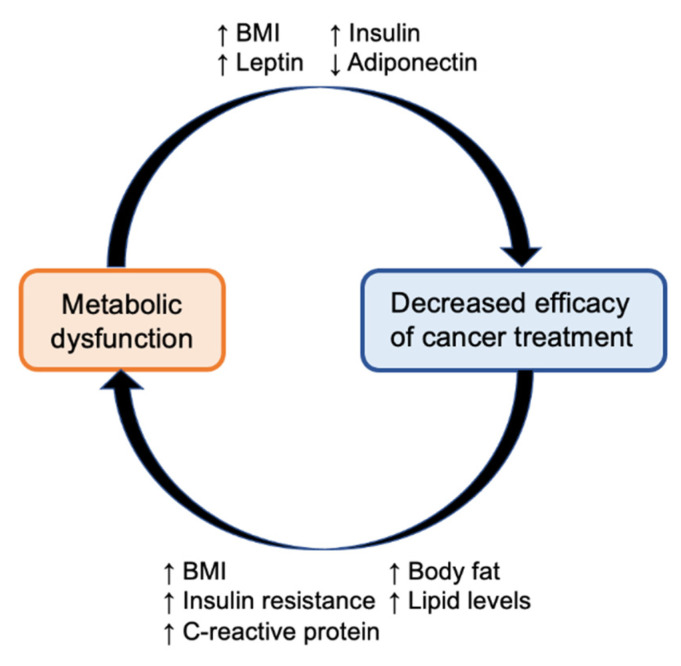
The paradoxical cycle of metabolic perturbation and treatment resistance: metabolic dysfunction, through various factors such as body mass index (BMI), insulin resistance, and altered levels of leptin and adiponectin, limits the efficacy of cancer treatment. Cancer treatments such as chemotherapy can worsen metabolic dysfunction, which in turn further limits treatment efficacy.

**Figure 2 nutrients-14-03542-f002:**
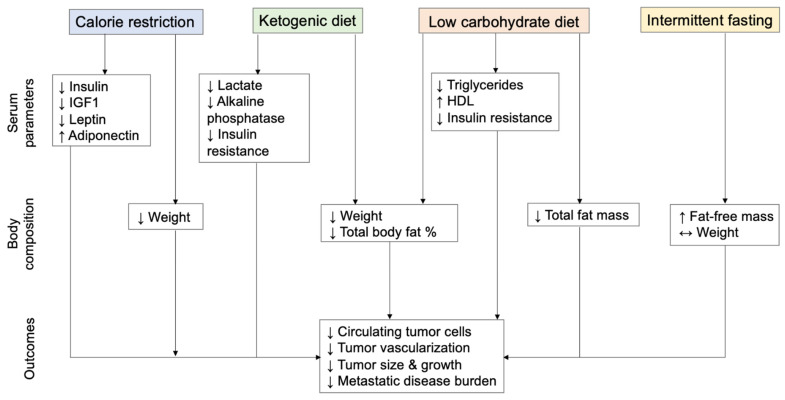
Effects of dietary strategies on metabolic parameters, body composition, tumor characteristics, and cancer outcomes in metastatic cancer.

**Table 2 nutrients-14-03542-t002:** Summary of clinical studies testing insulin-lowering dietary strategies that included patients with advanced or metastatic cancer.

Author	Year	N	Study Type	Site	Intervention	Findings
Khodabakhshi [58,59,60]	2020, 2021	80	RCT	Breast, any stage	Eucaloric KD (6% calories from carbohydrates, 19% from protein, 20% from medium-chain triglycerides, 55% from fat) × 90 days	↓BMI, ↓weight, ↓fat%, ↓fasting glucose, ↓insulin, ↑QOL, no difference in response rate in metastatic patients
Freedland [51]	2019, 2021	42	RCT	Prostate, any stage	≤20 g carbohydrate/day diet + walking (≥30 min for ≥5 days/week) × 6 months	↓weight, ↓fat mass, ↓% body fat, ↓insulin resistance, ↓hemoglobin A1c, ↑HDL, ↓TG, ↓HDL, no differences in PSA
Tan-Shalaby [61]	2016	17	Single-arm safety/feasibility study	Any site, advanced stage	20–40 g carbohydrates/day diet × 16 weeks	↓weight, ↑QOL, no unsafe adverse events, 36% achieved SD/PR, compliance was difficult
Fine [62]	2012	12	Single-arm safety/feasibility study	Any site, advanced stage	Carbohydrate dietary restriction to 5% of total kilocalories × 28 days	↓weight, no unsafe adverse effects, 42% achieved SD/PR; extent of ketosis correlated with response
Schmidt [63]	2011	16	Single-arm feasibility study	Any site, advanced stage	KD (<70 g carbohydrates/day, <20 g carbohydrates/meal)	Only 5/16 patients completed KD × 3 months, others discontinued due to difficult adherence or PD, mixed QOL changes, no severe adverse effects
Caffa [41]	2020	36	2 safety/feasibility studies	Breast, any stage	Periodic 5-day FMD every 4 weeks	↑fat-free mass, ↓fat mass, ↓blood glucose, ↓serum IGF1, ↓leptin, ↓C-peptide
Dorff [64]	2016	20	Single-arm safety/feasibility study	Any site, any stage	Fasting for 24, 48, or 72 h before platinum-based chemotherapy × 2 cycles	↓IGF1 levels, non-significant trend toward less grade 3/4 neutropenia
Iyikesici [65]	2018	44	Retrospective	NSCLC, stage IV	Metabolically supported chemotherapy (fasting- and insulin-induced hypoglycemia, local hyperthermia, hyperbaric oxygen therapy) + KD	ORR 61%, median OS 42.9 months, median PFS 41 months, no significant toxicity or adverse events due to KD
Iyikesici [66]	2020	25	Retrospective	Pancreatic, stage IV	Metabolically supported chemotherapy + KD	Median OS 15.8 months, median PFS 12.9 months, no significant toxicity or adverse events due to KD

Abbreviations: BMI, body mass index; kcal, kilocalories; IGF1, insulin growth factor-1; KD, ketogenic diet; LCD, low-carbohydrate diet; NSCLC, non-small cell lung cancer; PD, progressive disease; PR, partial response; QOL, quality of life; RCT, randomized controlled trial; SD, stable disease; ↓, decreased; ↑, increased.

**Table 3 nutrients-14-03542-t003:** Summary of ongoing studies testing insulin-lowering dietary strategies in patients with metastatic cancer.

NCT Number	Cancer Type	Target Accrual	Intervention	Primary and Secondary Endpoints
03795493 [67]	Breast cancer	50	CR (50% measured energy requirements) + aerobic exercise × 48–72 h prior to chemotherapy × 6 cycles vs. usual care	Tumor size on CT and MRI, treatment side effects, quality of life, PFS and OS
05090358	Breast cancer	106	KD vs. LCD vs. SGLT2 inhibitor × 12 weeks	Grade 3/4 hyperglycemia-free rate, ORR, PFS, alpelisib adherence and discontinuation rate, changes in systemic hormones and metabolites related to glucose homeostasis, changes in body weight and composition, and QOL measures
ChiCTR1900024597 [68]	Breast cancer	518	KD + irinotecan vs. normal diet + irinotecan	ORR, sensitivity to irinotecan, PFS, OS, QOL, incidence of grade 3–4 adverse events
04316520	Renal cell carcinoma	20	KD × 1 year	KD tolerance and adverse events, compliance, PFS, OS
05119010	Renal cell carcinoma	60	Continuous KD vs. intermittent KD (15 days on/15 days off) vs. intermittent oral liquid ketone supplement vs. standard diet × 3 months	ORR, grade 3–4 adverse events, weight, albuminemia/prealbuminemia, CRP levels, sarcopenia, QOL, PFS, OS
04631445	Pancreatic cancer	40	KD + triplet chemotherapy vs. normal diet + triplet chemotherapy	PFS, ORR, disease control rate, cancer biomarkers, BMI, insulin, A1c, serum metabolites, QOL
04708860	Breast cancer	30	Prolonged nightly fasting (13 h) × 12 weeks	Feasibility, adherence, metabolic biomarkers, QOL
04387084	Skin malignancies	16	Short-term fasting × 48 h prior to and 24 h after immunotherapy	Safety/feasibility, adherence, adverse events, ORR, immune-related toxicity, QOL, fasting-related biomarkers, immune biomarkers
02710721	Prostate cancer	60	60-h modified fasting (36 h prior and 24 h after chemotherapy) vs. Mediterranean diet	QOL, differential blood counts, chemotherapy-related adverse effects

Abbreviations: PFS, progression-free survival; OS, overall survival; ORR, overall response rate, QOL, quality of life; CRP, C-reactive protein; BMI, body mass index; A1c, hemoglobin A1c.

## Data Availability

Not applicable.

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
