# Peer review of "Insulin-Lowering Diets in Metastatic Cancer"

_nutrients, 2022, doi:10.3390/nu14173542_

Round 1
Reviewer 1 Report
This is a well written review of proposed mechanisms of insulin -lowering diets use in metastatic cancer. The summary and future directions are clear and supported by the published information reviewed.
Author Response
We thank the reviewer for this comment.
Reviewer 2 Report
This is a very interesting review of studies investigating the effects of insulin-lowering diets on metastatic cancer in both animal models and human clinical trials. The manuscript is clear and well organised. However, to improve the quality and impact of this paper for readers I suggest to describe more precisely the methods followed for the articles search and selection process according to the PRISMA statements (Preferred Reporting Items for Systematic Reviews and Meta-Analyses) (J Clin Epidemiol. 2009 62(10):e1-34. doi: 10.1016/j.jclinepi.2009.06.006). In this case, the review became a “systematic review” which, in my opinion, make the results more complete and reproducible, and useful clinicians, policy makers, and other users.
I would also suggest adding a summary table on the effects of insulin-lowering diets on animal models.
Author Response
To improve the quality and impact of this paper for readers I suggest to describe more precisely the methods followed for the articles search and selection process according to the PRISMA statements (Preferred Reporting Items for Systematic Reviews and Meta-Analyses) (J Clin Epidemiol. 2009 62(10):e1-34. doi: 10.1016/j.jclinepi.2009.06.006). In this case, the review became a “systematic review” which, in my opinion, make the results more complete and reproducible, and to useful clinicians, policy makers, and other users.
We thank the reviewer for this comment. This review was written as a narrative review rather than a systematic review. Since this is a narrow topic with limited evidence in both the preclinical and clinical settings, our intent was to describe the available literature and suggest future directions for conduct of clinical trials addressing the benefit of insulin-lowering diets in metastatic cancer. We chose not to write a systematic review and thus did not design our search according to PRISMA statements a priori for several reasons: 1) we felt that use of explicit criteria to include/exclude studies would even more significantly limit the studies we could discuss; and 2) we did not intend to extract or synthesize the findings given the significant heterogeneity between studies in terms of conduct and patient populations, since we included studies of all types of cancers in the metastatic setting. However, we appreciate the reviewer’s point, and have made the type of review clearer in the abstract text as follows (page 1):
“In this narrative review, we searched the published literature for studies that tested various insulin-lowering diets in metastatic cancer in preclinical and clinical settings.”
I would also suggest adding a summary table on the effects of insulin-lowering diets on animal models.
A summary table (Table 1) on the effects of insulin-lowering diets on animal models has now been included in the manuscript, and the table numbers for the subsequent tables (Tables 2 and 3) have been updated accordingly in both the text and table legends.
Reviewer 3 Report
This review article concerns the therapeutic dietary possibilities to support treatment of patients with advanced and metastatic cancer. Insulin-lowering dietary strategies such as calorie restriction (CR), low-carbohydrate or ketogenic diets (KD), and intermittent fasting (IF) can be very beneficial to attenuate cancer cells. A total of 23 studies were identified and KD, or IF strategies which appear to be safe and feasible.
In my opinion modulation of cancer cells metabolism is a very good idea to improve therapy new clinical trials in this field will be very important.
My suggestions:
1. Figures description should be under the figures.
2. Figure 1 should me more informative, at least at the level of figure description.
3. Interpunction in the textcan be corrected.
Author Response
We thank the reviewer for these comments.
- Figures description should be under the figures.
Figure descriptions are now under the figures.
- Figure 1 should me more informative, at least at the level of figure description.
The Figure 1 description has been expanded to be more informative.
- Interpunction in the text can be corrected.
We have reviewed the entirety of the text and corrected any punctuation where needed.